# Land-atmosphere interactions in sub-polar and alpine climates in the CORDEX FPS LUCAS models: II. The role of changing vegetation

Priscilla A. Mooney[1], Diana Rechid[2], Edouard L. Davin[3], Eleni Katragkou[4], Natalie de Noblet-Ducoudré[5], Marcus Breil[6], Rita M. Cardoso[7], Anne Sophie Daloz[8], Peter Hoffmann[2], Daniela C.A. Lima[7], Ronny Meier[3], Pedro M.M. Soares[7], Giannis Sofiadis[4], Susanna Strada[9], Gustav Strandberg[10], Merja H. Toelle[11] and Marianne T. Lund[8].

[1]NORCE Norwegian Research Centre, Bjerknes Centre for Climate Research, Bergen, Norway
[2]Climate Service Center Germany (GERICS), Helmholtz-Zentrum Hereon, Hamburg, Germany
[3]Wyss Academy for Nature, Climate and Environmental Physics, Oeschger Centre for Climate Change Research, University of Bern, Bern, Switzerland
[4]Department of Meteorology and Climatology, School of Geology, Aristotle University of Thessaloniki, Thessaloniki, Greece
[5]Laboratoire des Sciences du Climat et de l'environnement, Paris, France
[6]Institute for Meteorology and Climate Research, Karlsruhe Institute of Technology, Karlsruhe, Germany
[7]Instituto Dom Luiz, Faculdade de Ciências da Universidade de Lisboa, 1749-016 Lisboa, Portugal
[8]CICERO Center for International Climate Research, Oslo, Norway
[9]International Center for Theoretical Physics, Trieste, Italy
[10]Swedish Meteorological and Hydrological Institute, Norrkoping, Sweden
[11]Center for Environmental Systems Research, University of Kassel, Germany

*Correspondence to*: *P.A. Mooney (prmo@norceresearch.no)*

**Abstract.** Land cover in sub-polar and alpine regions of northern and eastern Europe have already begun changing due to natural and anthropogenic changes such as afforestation. This will impact the regional climate and hydrology upon which societies in these regions are highly reliant. This study aims to identify the impacts of afforestation/reforestation (hereafter afforestation) on snow and the snow-albedo effect, and highlight potential improvements for future model development. The study uses an ensemble of nine regional climate models for two different idealised experiments covering a 30-year period; one experiment replaces most land cover in Europe with forest while the other experiment replaces all forested areas with grass. The ensemble consists of nine regional climate models composed of different combinations of five regional atmospheric models and six land surface models. Results show that afforestation reduces the snow-albedo sensitivity index and enhances snow melt. While the direction of change is robustly modelled, there is still uncertainty in the magnitude of change. Greatest differences between models emerge in the snowmelt season. One regional climate model uses different land surface models which shows consistent changes between the three simulations during the accumulation period but differs in the snowmelt season. Together these results point to the need for further model development in representing both grass-snow and forest-snow interactions during the snowmelt season. Pathways to accomplishing this include 1) a more sophisticated representation of forest structure, 2) kilometer scale simulations, and 3) more observational studies on vegetation-snow interactions in Northern Europe.

## 1 Introduction

Interactions between the land surface and the atmosphere in sub-polar and alpine climates occur largely through the snow albedo effect in winter and spring. These interactions strongly influence the regional climate and any change to either land cover or snow cover in these regions will alter the regional climate (IPCC, 2019; Cherubini et al., 2018; Bender et al., 2020). Importantly, changes to the land surface, such as afforestation, also alter the snow cover (Mooney et al., 2021).

Land cover is undergoing rapid change in many parts of the world, including in the sub-polar and alpine regions. Some of this change is a natural response to climate change, e.g., forest fires (Wang et al., 2020) and "greening of the Arctic" (Myers-Smith et al., 2020). Other changes to the land surface are a more direct human influence such as afforestation (Mooney et al., 2021). The impact of these perturbations to the land surface on the regional climate and hydrology can have considerable consequences for society. In these regions, many communities rely on snow for water resources, tourism, energy, and recreation (Framstad et al. 2009; Duncker et al., 2012). They are also vulnerable to snow-related hazards such as flooding and avalanches (Abermann et al., 2019).

Many observation-based studies have assessed the impact of forests on snow accumulation and loss compared to open sites such as grasslands (e.g., Golding and Swanson, 1978; Essery et al., 2003; Varhola et al., 2010; Lundquist et al., 2013). These studies and references therein have shown that the impacts of forests on snow accumulation and ablation are dependent on vegetation structure, local climate, topography, and aspect. Forests can reduce snowpack compared to grasslands through canopy interception and emitting longwave radiation. Conversely, forests can enhance snowpack by shading it from solar radiation, and sheltering it from strong winds (Varhola et al., 2010).

One factor that influences the magnitude of snowpack reduction through canopy interception is forest density. For dense forests, canopy interception can lead to losses in snowpack that exceed 60% of the total annual snowpack (Hedstrom and Pomeroy, 1998). This loss in snowpack during the accumulation period can be offset by the canopy's shading of the snowpack from solar radiation and strong winds in the snowmelt season, if this is the dominant mechanism for snow loss (Lundquist et al., 2013). However, the loss of snowpack during the snowmelt season could be dominated by increased longwave radiation instead if winter temperatures (DJF) exceed -1°C (Lundquist et al., 2013).

Representing these highly complex interactions between forests and snow cover poses a challenge for both global and regional climate modelling (Mudryk et al., 2020). In regional climate models, key processes for vegetation-snow interactions are simulated by both the atmospheric model and the land surface model. During the accumulation phase, atmospheric processes most strongly influence snowpack characteristics, but during the snowmelt season, land surface processes are most influential. Various studies (e.g., Essery et al., 2003; Mudryk et al., 2020; Mooney et al., 2020) have demonstrated that while climate models have become more sophisticated in their representation of vegetation-snow interactions and have improved in their ability to simulate snow cover, there are still deficiencies in the simulation of snow amount.

In Daloz et al., (in review; hereafter Part I), we have shown that deficiencies in the simulation of the snow-albedo climate forcing, a key land-atmosphere interaction in sub-polar and alpine climates, are greatest during the snowmelt period for

different regional climate models participating in the World Climate Research Programme's (WCRP) Coordinated Regional Climate Downscaling Experiment (CORDEX) endorsed Flagship Pilot Study (FPS) Land Use and Climate Across Scales

(LUCAS; Rechid *et al*., 2017), hereafter called CORDEX FPS LUCAS.

These model deficiencies combined with limited observations means much remains unknown about the impact of afforestation on the regional climate system and snowpack characteristics in sub-polar and alpine climates of northern and eastern Europe. This study will address this issue and further focus future model development for vegetation-snow interactions using an ensemble of nine CORDEX FPS LUCAS simulations for two different and extreme land cover changes. While Part I used

simulations with a realistic land cover map, this study (Part II) uses simulations with idealised land cover maps that cover most of Europe with forest in one experiment and grass in the other.

The aims of this study are (1) to identify robust impacts of afforestation on the snow-albedo effect, snow variables, and a selection of societally relevant metrics, and (2) highlight required improvements for model development in these regions. This study is the first to investigate land atmosphere interactions with a focus on snow variables in high latitude regions by using

an ensemble of regional climate models with idealised land cover scenarios specifically designed to assess the impact of afforestation in Europe. In doing so, this study will provide one of the most robust assessments of the impact of afforestation on snowpack in northern and eastern Europe to date. A description of the methodology can be found in the next section, and the results are presented in section 3. These results are further discussed in section 4 and the conclusions are presented in section 5.

**2 Methodology**

**2.1 *CORDEX FPS LUCAS Experiments***

Simulations in the CORDEX FPS LUCAS were performed for three different types of experiments: EVAL, FOREST, and GRASS. All simulations use a grid spacing of 0.44°, cover the period 1986-2015, and use the standard EURO-CORDEX domain (Jacob et al., 2014). Boundary and initial conditions for all simulations were derived from the European Centre for

Medium range Weather Forecasting's Interim reanalysis (ERA-Interim; Dee et al., 2011). The difference between the EVAL, FOREST, and GRASS experiments lies in the land cover maps. Simulations for the EVAL experiment use the present-day land cover map specific to each regional climate model (RCM). The EVAL simulations are the control simulations and have been used to evaluate the performance of the different RCMs in Part I (Daloz *et al*. (in review)) and by Davin et al., (2020) and Sofiadis et al. (2021). The FOREST and GRASS experiments, which are the focus of this paper, use idealised land cover

maps (see Figure 1 for an example) that are designed to represent the theoretical maximum of forest and grass coverage. These idealised land cover maps are derived from a MODIS-based present-day land cover map. From this map, the fractional coverage of forest is expanded until it covers 100% of non-bare soil ground. The FOREST map conserves the ratio of tree types (i.e., broadleaf vs needle leaf and deciduous vs evergreen) found in the MODIS-based land cover map. The GRASS land

cover map was developed in the same way as the FOREST land cover map. A more comprehensive description of the land cover maps and conversion rules can be found in (Davin et al., 2020).

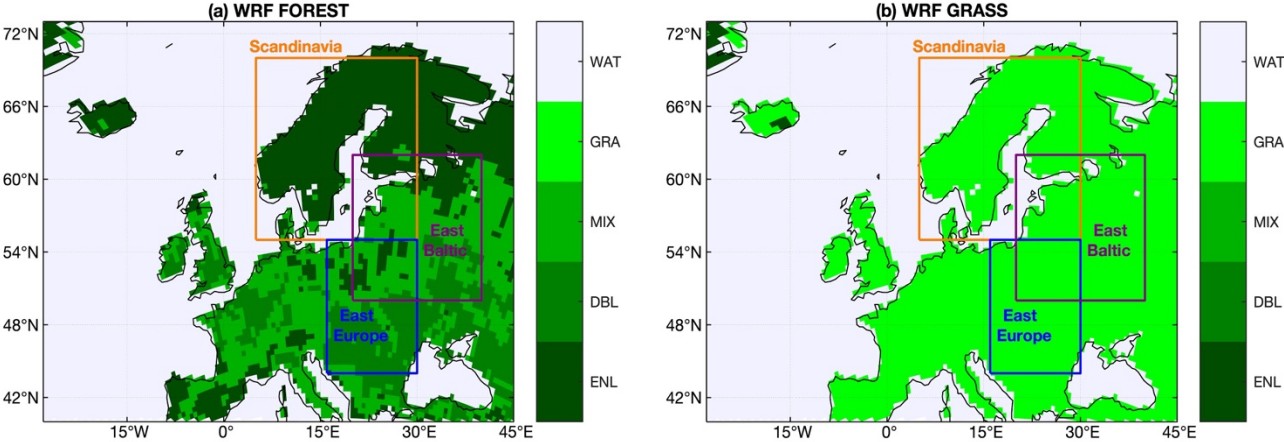

**Figure 1:** The three regions considered in the analysis: Scandinavia, East Baltic, East Europe. Also shown are the land cover maps for the FOREST and GRASS experiments. The colour bars indicate land cover type where WAT means water, GRA means grass, MIX means mixed forests, DBL means deciduous broadleaf, and ENL means evergreen Needleleaf.

### 2.2 *CORDEX FPS LUCAS models*

This study uses nine RCMs composed of different combinations of five regional atmospheric models and six LSMs. The combinations are shown in Table 1 which also specifies the model versions, key references for each model and their representation of snow-vegetation interactions. The ensemble consists of two regional models (WRF and CCLM) that use multiple LSMs allowing the analysis to isolate the impact of the LSMs on the results. Uniquely, the ensemble consists of two WRF-NoahMP simulations that differ only by their representation of convection and planetary boundary layer processes. Hereafter, each of these combinations will be considered as different RCMs as they differ in the way they represent different atmospheric and land surface processes.

Data for all snow variables was available for WRFc-NoahMP, CCLM-CLM5.0, WRFa-NoahMP, RCA, WRFb-CLM4.0. However, REMO-iMOVE, RegCMa-CLM4.5, and CCLM-TERRA did not have data for snow depth while CCLM-VEG3D could only provide a binary number of 0 and 1 for snow cover fraction. Snow depth for REMO-iMOVE, RegCMa-CLM4.5, and CCLM-TERRA was calculated from snow water equivalent using a constant value of 312 kg/m$^3$ for the bulk density as described in Sturm *et al*. (2010).

### 2.2.1 Snow Vegetation Interactions

Below we provide a short description of the snow-vegetation interactions in the different land surface models. A more comprehensive description for each model can be found in the references listed in Table 1.1 for each model.

*Canopy interception:* All models except RCA account for canopy interception of precipitaton. WRFa-NoahMP, WRFc-
NoahMP, CCLM-CLM5.0, CCLM-Veg3D, and REMO-iMOVE use a separate layer for the vegetation canopy which allows for both liquid water and ice to be intercepted by the vegetation canopy. CCLM-TERRA also accounts for interception of rain and snow using separate reservoirs for each. While WRFb-CLM4.0 and RegCM-CLM4.5 also use a separate layer for the canopy canopy, the interception of precipitation does not distinguish between rain and snow.

*Turbulent transfer under canopy:* In WRFa-NoahMP, WRFc-NoahMP, CCLM-Veg3D, CCLM-TERRA, and REMO-iMOVE,
turbulent fluxes between snow, vegetation, and air use Monin-Obukhov Similarity Theory (MOST) stability functions to calculate aerodynamic resistance with respect to the displacement and roughness lengths of the canopy. In RCA4, turbulent transfer within the canopy layer is parameterized following Choudhury and Monteith (1988). The canopy air state is connected to the canopy itself and to the surface beneath the canopy with separate aerodynamic resistances, respectively. In WRFb-CCLM4.0, CCLM-CLM5.0 and RegCMa-CLM4.5, aerodynamic resistances to heat/moisture transfer between the ground and
the canopy air account for the turbulent transfer via a coefficient that results from interpolation between values for dense canopy and bare soil (Zeng et al. 2005). The dense canopy turbulent transfer coefficient in Zeng et al. (2005) is modified from its original value of 0.004 to account for stability.

*Radiative Transfer:* WRFa-NoahMP, WRFc-NoahMP, WRFb-CLM4.0, CCLM-CLM5.0 and RegCMa-CLM4.5 use the two-stream approximation of Dickinson (1983) and Sellers (1985) for radiative transfer within vegetation canopies. CCLM-Veg3D
also uses a two stream approach but the radiative transfer is calculated after the approach of Ritter & Geleyn (1992). In RCA, the net radiation components are separated between the forest canopy and the forest floor. In this separation a sky view factor is applied which accounts for the degree of canopy closure (Verseghy et al., 1993) which is a function of LAI only for long-wave and diffuse short-wave radiation. In CCLM-TERRA, the radiative transfer equations are calculated in terms of upward and downward fluxes based on the two-stream methods. Surface radiation fluxes depend on surface albedo and temperature.
In REMO-iMOVE, there is no direct canopy radiative transfer, land surface influence is calculated for visible and near-infrared through albedo changes.

*Snow Albedo:* The surface albedo in WRFa-NoahMP, WRFc-NoahMP, and WRFb-CLM4.0 is a diagnostic produced from the ratio of total reflected shortwave radiation to total downward shortwave radiation. In RCA4, snow albedo for open-land snow is a prognostic variable and the albedo for snow in the forest is set constant to 0.5. In CCLM-Veg3D, snow albedo is a
prognostic quantity depending on the age of the snow. CCLM-TERRA has a time-dependent snow albedo introduced by an ageing-function with pre-given maximum and minimum snow albedo values which accounts for partial coverage of the soil surface by vegetation and snow for albedo.  In REMO-iMOVE, the snow albedo is calculated as a function of snow surface temperature and forest cover. The snow covered land surface albedo is a function of the snow albedo, the background albedo,

and the actual snow depth. In both CCLM-CLM5.0 and RegCMa-CLM4.5, snow albedo within each snow layer is simulated
with the Snow, Ice, and Aerosol Radiative Model (SNICAR), which incorporates a two-stream radiative transfer solution from
Toon et al. (1989) and considers snow age.

| Model Name | Institute ID | RCM | LSM | Snow - vegetation interaction |
|---|---|---|---|---|
| WRFa-NoahMP | **IDL** | WRF v3.8.1D (Powers et al., 2017) | NoahMP (Niu et al., 2011) | (Deardorff, 1978; Niu and Yang, 2007) |
| WRFc-NoahMP | **BCCR** | WRF v3.8.1 (Powers et al., 2017) | NoahMP (Niu et al., 2011) | (Deardorff, 1978; Niu and Yang, 2007) |
| WRFb-CLM4.0 | **AUTH** | WRF v3.8.1 (Powers et al., 2017) | CLM4.0 (Oleson et al., 2010) | (Wang and Zeng, 2009) |
| CCLM-CLM5.0 | **ETH** | Cosmo_5.0_clm9 (Sørland et al., 2021) | CLM5.0 (Lawrence et al., 2019) | (Wang and Zeng, 2009; Lawrence et al., 2019; van Kampenhout et al., 2017) |
| CCLM-VEG3D | **KIT** | Cosmo_5.0_clm9 (Sørland et al., 2021) | VEG3D (Braun and Schädler, 2005) | (Grabe 2002) |
| CCLM-TERRA | **JLU** | Cosmo_5.0_clm9 (Sørland et al., 2021) | TERRA-ML (Schrodin and Heise, 2002) | (Doms *et al.* 2013) |
| RegCMa-CLM4.5 | **ICTP** | RegCM v4.6 (Giorgi et al., 2012) | CLM4.5 (Oleson et al., 2013) | (Wang and Zeng, 2009) |
| REMO-iMOVE | **GERICS** | REMO2009 (Jacob et al., 2012) | iMOVE (Wilhelm et al., 2014) | (Roeckner *et al.*, 1996; Kotlarski, 2007) |
| RCA4 | **SMHI** | RCA4 (Strandberg et al. 2015) | Internal (Samuelsson et al., 2006) | (Samuelsson *et al.* 2015) |

**Table 1.1** A list of the RCMS, LSMs, and the model names used in this study. Also listed are the key references describing the models, and
the institutions that performed the simulations. Institution and model abbreviations are shown in Appendix A.

## 2.3 *Snow Albedo Sensitivity Index (SASI)*

The key interaction between the land and the atmosphere in sub-polar and alpine climates is through changes in surface albedo
during winter and spring. This study uses the SASI index (Xu and Dirmeyer, 2013) which is a measure of the climate forcing
from the snow-albedo effect. SASI has units of W/m$^2$ and is defined mathematically as:

$$SASI = SW\sigma(f_{sno})\Delta\alpha \ ,$$

where $SW$ is the incoming shortwave radiation, $\sigma(f_{sno})$ is the standard deviation of snow cover fraction over time, and $\Delta\alpha$ is the difference in surface albedo between the snow-covered surface and non-snow covered surface. In this study, $\Delta\alpha$ has a constant value of 0.5 for grass and 0.2 for forests as the values for this could not be obtained from the models. The albedo values used for snow covered grass and forest were 0.70 and 0.35 based on (Barlage *et al.*, 2005), while albedo values used for non-snow covered grass and forest were 0.2 and 0.15 based on (Myhre and Myhre, 2003). Although there are differences in the snow-covered values for different types of forests, i.e., deciduous vs evergreen, these differences were very small (0.35 vs 0.34). Previous studies on the impacts of afforestation in sub-polar climates, such as Mooney et al. (2021), have shown that such small changes in albedo have a negligible impact on the outcome. Based on this, the same value was used for all forests regardless of forest type. High SASI values of 10 W/m$^2$ or more indicate a strong radiative forcing from the snow-albedo effect.

### 2.4 *Start date of snowmelt season*

This study follows the definition of Xu and Dirmeyer (2011) to identify the start date for the snowmelt season. The start date for the snowmelt season is determined when the 5-day running mean of snow water equivalent falls to 80% of its peak value.

## 3 Results

### 3.1 SASI

Figure 2 shows the temporal evolution of SASI for the FOREST and GRASS simulations in Scandinavia, East Baltic, and East Europe for January to June. Values for July to December are excluded due to the lack of snow cover and/or low levels of incoming solar radiation. SASI is typically low in January since incoming solar radiation is low. As the season progresses, SASI values increase with increasing solar radiation until the snow starts melting. As the snow cover decreases, SASI values decrease and reach zero in most places by June. Consequently, the timing of snowmelt differs in the different regions due to latitudinal differences leading to different times for peak values of SASI.

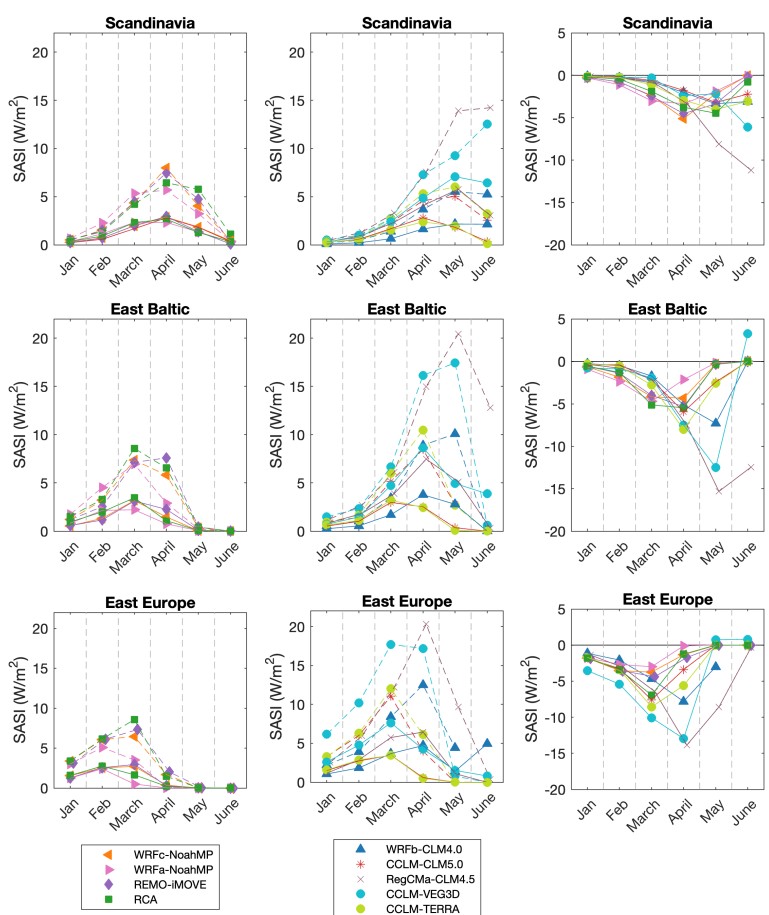

**Figure 2:** The first and second columns show the SASI values for GRASS (dashed lines) and FOREST (solid lines) experiments averaged over the three different regions shown in Figure 1. The nine simulations are divided into two different columns based on their values (high Vs low); the purpose of this separation is to ease interpretation of the results. The third column shows the impact of afforestation on SASI (FOREST – GRASS).

Figure 2 shows that GRASS simulations have higher SASI values than FOREST simulations meaning afforestation reduces the climate forcing from the snow-albedo effect. This is consistent with the findings of Davin et al. (2020) which also used the CORDEX FPS LUCAS models and showed that afforestation increased surface temperatures in all of these CORDEX FPS LUCAS models. Davin et al. (2020) also showed that afforestation increased net shortwave radiation and sensible heat considerably in these same models. This points to the decreased albedo from afforestation enhancing net shortwave radiation and leading to a positive response in surface air temperature to afforestation. In light of this, the lower SASI values for the FOREST simulations compared to the GRASS values can be primarily attributed to the difference in $\Delta\alpha$ which is 0.5 for GRASS and 0.2 for FOREST. Generally, afforestation does not impact the timing of the maximum value in SASI.

Four RCMs (WRFa-NoahMP, WRFc-NoahMP, REMO-iMOVE, and RCA) produce similar SASI values to each other for the GRASS experiment, and also for the FOREST experiment. The other RCMs simulate considerably different values for SASI in the GRASS experiment, with the largest differences appearing in the snowmelt season (April-June) when the SASI for some simulations can be 2-3 times larger than SASI values for other simulations. This is also evident in the FOREST experiment. It is important to note here that results for CCLM-VEG3D may arise from the use of a binary number (0 or 1) for snow cover fraction. The next subsection presents the impact of afforestation on snow water equivalent and snow cover which are key variables for SASI.

### 3.2 *Snow water equivalent and snow depth*

Snow water equivalent and snow depth are considered together as there is a relationship between these quantities and three of the models provide only snow water equivalent from which snow depth is derived by using a constant density value of 312 kg/m$^3$.

Figure 3 shows the difference between the FOREST and GRASS experiments for snow water equivalent for the nine different models. Only differences that are statistically significant at the 95% confidence level using the student t-test are shown. Four of the models show that afforestation reduces snow water equivalent in all months and one model shows that afforestation increases snow water equivalent in all months. The remaining four models show more spatial variability in both magnitude and sign of change.

A summary of Figure 3 is presented in Figure 4 which shows the spatial variability and mean in the difference between FOREST and GRASS experiments. In Scandinavia, most RCMs show that afforestation reduces snow water equivalent with modest decreases and little spatial variability during the accumulation phase, but large decreases and large spatial variability during the snowmelt season. Four RCMs (WRFa-NoahMP, WRFb-CLM, WRFc-NoahMP, and CCLM-TERRA) show that afforestation increases snow water equivalent during the accumulation period. Three of these RCMs show that afforestation also leads to higher values of snow water equivalent during the snowmelt season; WRFb-CLM shows that afforestation decreases the snow water equivalent during the melt season. The results for East Baltic are similar to Scandinavia despite the difference in forest type; Scandinavia has predominantly evergreen needleleaf forests, while East Baltic is dominated by mixed forest with considerable areas of deciduous broadleaf and evergreen needleleaf forests. In both Scandinavia and East Baltic, models display a greater spatial variability during the snowmelt season than during the accumulation period. Only small differences for a few models are shown in East Europe. However, values for snow water equivalent are smaller in this region compared to the others. The results for snow depth are not shown here as they are very similar to the results for snow water equivalent.

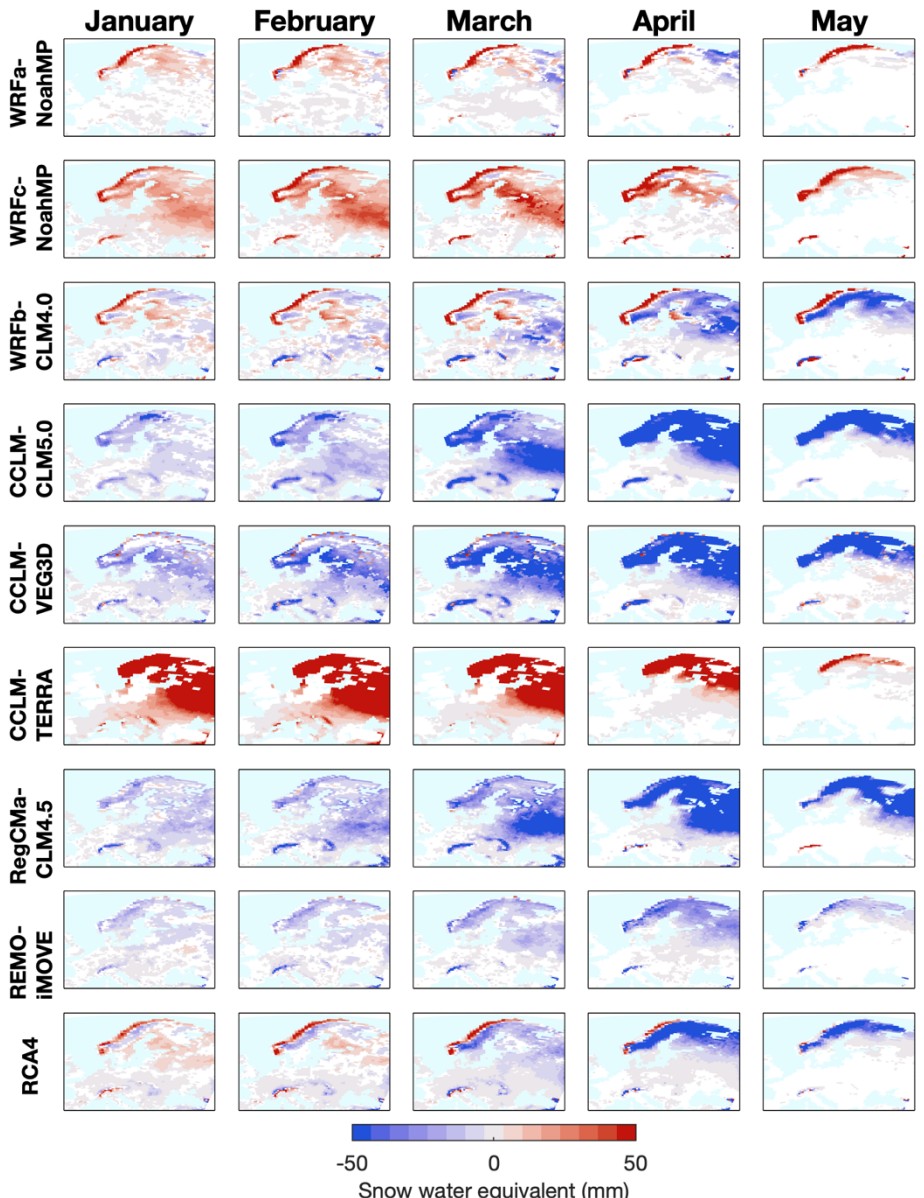

**Figure 3:** Impact of afforestation (FOREST-GRASS) on snow water equivalent (SWE) simulated by the CORDEX FPS LUCAS models. White spaces show grid boxes that are not statistically significant at the 95% confidence level.

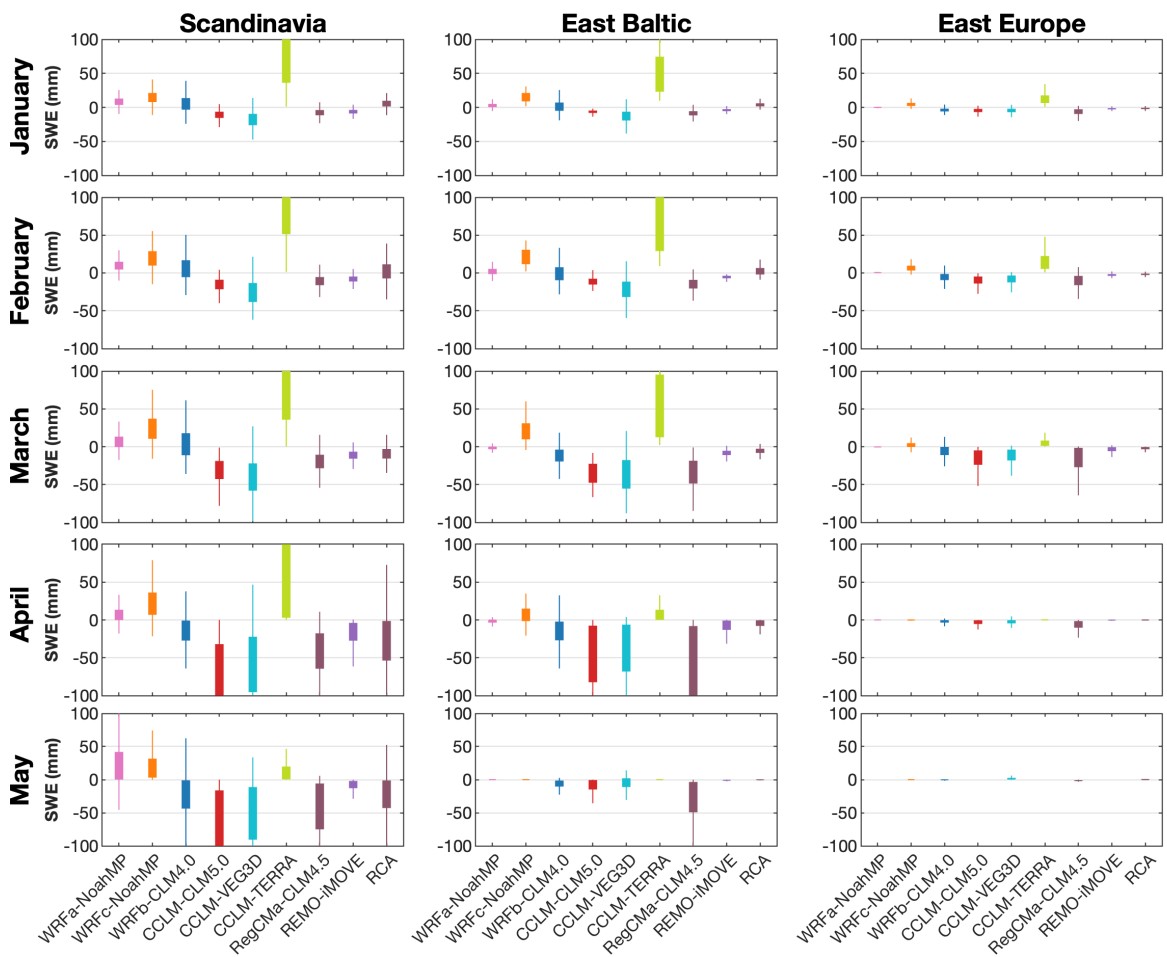

**Figure 4:** Impact of afforestation (FOREST-GRASS) on snow water equivalent (SWE) simulated by the LUCAS models. The box plots indicate the spatial variability in the difference between the FOREST and GRASS experiments for each month from January to May (see y-axis). Only differences that are statistically significant at the 95% confidence level are considered. Statistical significance was determined using the student t-test.

### 3.3 *Snow Cover*

Figure 5 shows the spatial variability in the difference of snow cover fraction between the FOREST and GRASS experiments. As in Figures 3 and 4, only differences that are statistically significant at the 95% confidence level are considered. Most notable in these results is the strong effect of afforestation and large spatial variability demonstrated by the three CCLM models during the snowmelt season. All results show a reduction in snow cover due to afforestation in the snowmelt season with little impact evident during the accumulation period. All three regions show some impact of afforestation on snow cover.

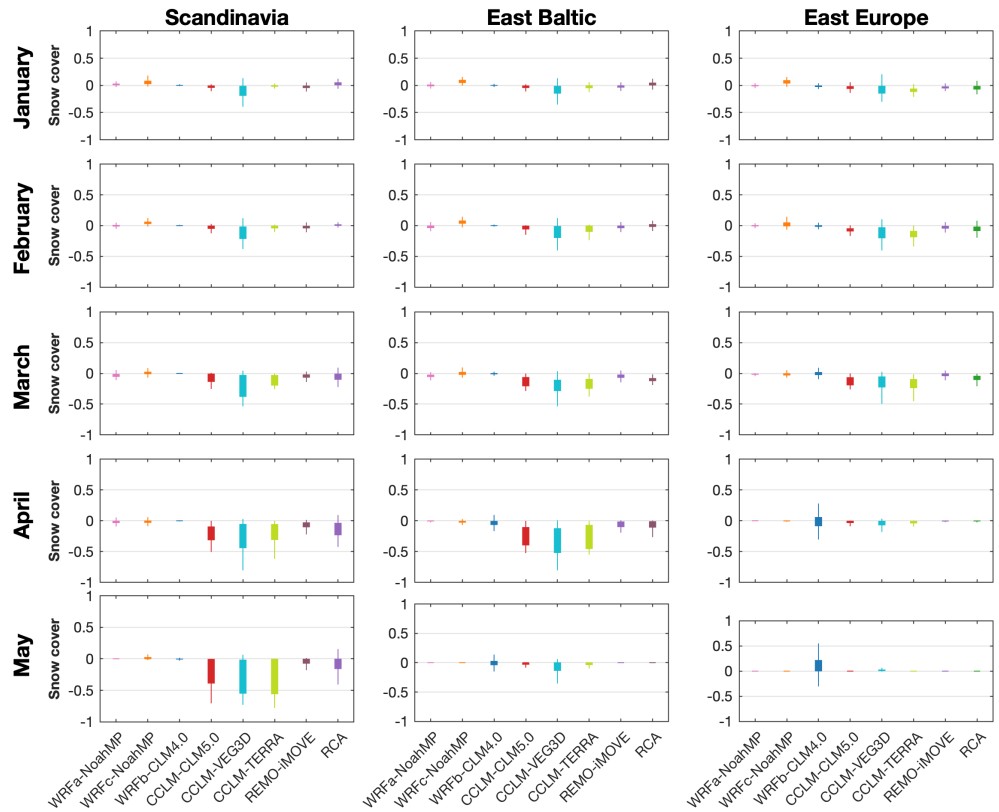

**Figure 5:** Impact of afforestation (FOREST-GRASS) on snow cover fraction simulated by the CORDEX FPS LUCAS models. The box plots indicate the spatial variability in the difference between the FOREST and GRASS experiments. Only differences that are statistically significant at the 95% confidence level are considered. Statistical significance was determined using the student t-test.

## 3.4 Snow Days

Figure 6 shows the impact of afforestation on the number of snow days. Snow days are defined as days when snow depth exceeds 0.1 m and the number of snow days is indicative of the length of the snow season. Four RCMs show that afforestation increases the number of snow days while five RCMs show that afforestation decreases the number of snow days. Three of the four RCMs showing an increase are from the WRF modelling system; three of the five RCMs showing a decrease are from the CCLM model. Both the WRF and CCLM ensembles consist of different LSMs. This suggests that differences in the representation of atmospheric processes are largely responsible for this conflicting result. Nonetheless, there are differences in the magnitude of the response to afforestation within the WRF and CCLM ensembles, suggesting that land surface processes are also important.

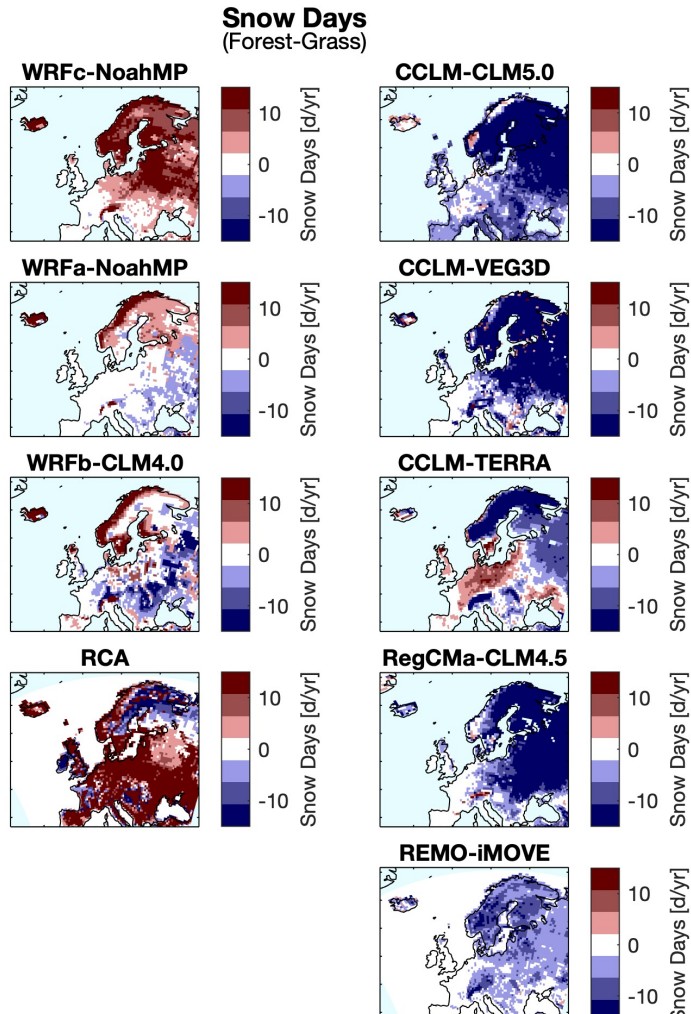

**Figure 6:** Impact of afforestation (FOREST-GRASS) on the number of snow days in the season, an indicator for the length of the snow season.

## 3.5 Snowmelt

Figure 7 shows the impact of afforestation on the start of the snowmelt season. The start of the snowmelt season is determined when the 5-day moving mean of snow water equivalent reaches 80% of the season maximum in the 5-day moving mean of snow water equivalent. In general, the results of Figure 7 show that afforestation tends to delay the onset of snowmelt. This is most evident in Scandinavia and the East Baltic regions. In the East Europe region, the mean value for most RCMs also shows a delay in the start of the snowmelt season. However, there is large spatial variability and two RCMs (REMO-iMOVE and WRFa-NoahMP) have a mean value greater than zero suggesting an earlier start of the snowmelt season.

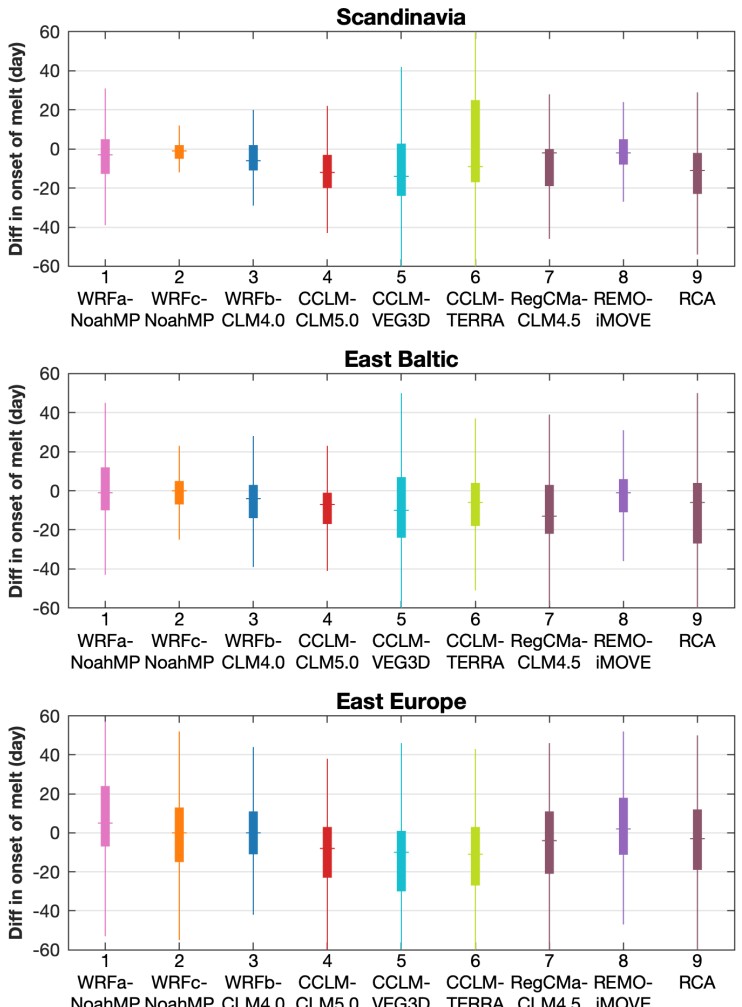

**Figure 7:** Impact of afforestation (FOREST - GRASS) on the start of the snowmelt season in the three different regions shown in Figure 1. The snowmelt season starts when the 5-day moving mean of snow water equivalent reaches 80% of the maximum value of the 5-day moving mean in the season.

## 4 Discussion

As highlighted in the companion paper (Daloz *et al*., in review) and other studies (e.g. Matiu et al., 2020), regional climate models have substantial difficulties in simulating snow related processes and variables. This study highlights the need for further model development in the representation of vegetation-snow interactions during the snowmelt season. This is evident from disagreements between models on the magnitude and sign of change arising from afforestation for some of the analyses. It is further evident in the disagreement between model results and observations shown in the companion paper of Daloz *et al.* (in review), especially during the snowmelt season. Model improvements in the representation of vegetation-snow interactions

can substantially reduce known biases in regional climate simulations for other climate variables in northern and eastern Europe (Mooney et al., 2013; Katragkou et al., 2015). Such improvements would increase confidence in climate change projections for these regions.

Observational studies using paired-site experiments of forests and open spaces, such as grass, have shown that afforestation generally decreases snow accumulation and lowers melt rates (Varhola et al., 2010, and references therein). However, the processes behind these results are very complex and highly variable depending on multiple factors that have led to conflicting results (Lundquist et al., 2013). While all models struggle to reproduce these complexities there are some robust findings here. Models show that afforestation has the greatest impact during the snowmelt season and there is good agreement between the models in simulating the impacts on snow cover. This is consistent with other international studies assessing the ability of climate and land surface models to simulate snow cover (Essery et al., 2009; Mudryk et al., 2020; Krinner et al., 2018). However, there is less agreement in the magnitude of changes during the snowmelt season when afforestation impacts are greatest. Simulating snow-vegetation interactions during snowmelt is a known challenge for models (Krinner et al., 2018). The models also showed good agreement in simulating the impact of afforestation on the onset of the snowmelt season although there was disagreement on the magnitude of change. Disagreement was also found on the impact of afforestation on snow water equivalent. This may be related to the known deficiencies in climate models to simulate snow mass variables, such as snow water equivalent, highlighted in previous studies (Thackeray et al., 2019; Mudryk et al., 2020).

Societies in many sub-polar and mountainous regions of the world depend on snow accumulation and snowmelt for a myriad of social and economic activities e.g., water resources and winter tourism. Indeed, these regions are also vulnerable to flooding and avalanches. Regardless of the sign of change, if the impact of afforestation or deforestation on snow accumulation and/or melt is sufficiently large, communities in these regions will be impacted by afforestation. Certain ecological species are also highly dependent on snow, and if afforestation mitigates some of the snow melt expected in a warmer world, then afforestation could help conservation efforts by offering potential sites for "climate change refugia" in a warmer world. Together these points, highlight the societal and ecological need for better information on the impact of afforestation in sub-polar and alpine regions, some of which are already undergoing afforestation.

**5 Conclusions**

In this study, we used an ensemble of RCMs to investigate the impact of afforestation during January-June on the climate forcing due to the snow-albedo effect, which is a key land-atmosphere interaction in sub-polar and alpine climates. The study showed that afforestation decreases the snow-albedo climate forcing. This is largely due to changes in surface albedo. While models agreed on the sign of change, there was disagreement in the magnitude of the impact of afforestation on SASI. Results also showed that there was no impact on the timing of the peak value of SASI which generally occurs in March or April depending on the region. Our study also showed that there was a large spread in the values for both the FOREST and GRASS simulations, suggesting that model improvements are required for both grass-snow and forest-snow interactions.

The study also examined the impact of afforestation on snow water equivalent, snow depth, and snow cover fraction. Most models show that afforestation has a smaller impact in January and February when snow is generally accumulating than in March, April and May when snow is melting. Most models showed that afforestation reduced snow water equivalent, snow depth and snow cover fraction in March, April and May when snow is typically melting. However, the models do disagree on the magnitude of the change. This indicates that afforestation enhances snowmelt with little to no impact on snow accumulation. Afforestation was also shown to generally delay the start of the snowmelt season. Analysis of the impact of afforestation on the number of snow days was inconclusive with four models showing increases and five models showing clear decreases.

The main limitations of this study are 1) coarse model resolution, 2) inadequate model representation of complex forest-snow interactions, and 3) lack of forest-snow observations. The coarse spatial resolution in this study limits the ability of all models to adequately represent essential atmospheric processes such as precipitation and key land surface processes and characteristics, e.g., elevation and canopy-snow interactions. Another limitation is the simplistic representation of forest-snow interactions even in the most sophisticated models. For example, most models do not consider the role of forest density in forest-snow interactions, even though observation-based studies have shown the importance of this forest characteristic, and there are well known differences in forest density between managed and natural forests. Another well known deficiency in regional models is their inability to represent windblown snowdrift, which is an important factor for quantifying the snow-albedo effects of afforestation. Finally, the study's ability to determine which model or models correctly represent vegetation-snow processes is severely hampered by the lack of high-quality observations of surface energy and moisture fluxes in forests and grasslands in these regions, particularly in Scandinavia.

These limitations highlight the need for future developments in land surface models to focus on a more sophisticated representation of forest-snow interactions such as the impact of forest type, density, and atmospheric temperatures on both snowmelt and snow accumulation. Indeed, such development would also enhance the performance of regional climate models in these regions.

Future studies should consider using kilometer-scale resolutions as computational resources are becoming more affordable. This would better represent important atmospheric processes and aspects of the land surface such as precipitation processes, and mountainous terrain. This is particularly important in Scandinavia where models in this study show large differences in snow water equivalent. Convection permitting models would not only improve the amounts of precipitation but also its classification into rain and snow would be based on microphysical processes instead of the threshold-based approaches used in coarser models. The next two phases of CORDEX FPS LUCAS will be implemented at higher resolutions with the third phase applying kilometre-scale resolutions. This will provide additional knowledge and insights on this important topic.

Future studies could also consider the impact of dynamic vegetation modelling on the snow-albedo feedback. Previous studies such as Cook et al. (2008) have shown that dynamic vegetation in climate models can be an important amplifier of the snow-albedo feedback. Such analysis was not possible in this study as most models did not have this capability. Future studies should examine this when more land surface models have developed this capacity.

There is also a need for more observational work on vegetation-snow interactions, particularly in northern Europe. A number of observational studies have been conducted in Canada, Russia and the United States of America but only a few have been carried out in Northern Europe. Existing observational studies show that snow-vegetation interactions depend on numerous factors, including elevation and climate. This implies that the results from studies in other regions may not necessarily apply to Northern Europe. Such observations would advance our understanding of vegetation-snow interactions, support model

evaluation, improve model development, and reduce uncertainty in future climate projections.

## Appendix A – Abbreviations

| Institution | |
|---|---|
| *AUTH* | Aristotle University of Thessaloniki |
| *BCCR* | Bjerknes Centre for Climate Research |
| *BTU* | Brandenburgische Technische Universität |
| *ETH* | Eidgenössische Technische Hochschule Zürich |
| *GERICS* | Climate Service Center Germany |
| *ICTP* | International Centre for Theoretical Physics |
| *IDL* | Instituto Amaro Da Costa |
| *JLU* | Justus-Liebig-Universität Gießen |
| *KIT* | Karlsruhe Institute of Technology |
| *SMHI* | Swedish Meteorological and Hydrological Institute |
| **Regional Climate Models** | |
| *CCLM* | Cosmo-CLM |
| *RCA* | Rossby Centre regional Atmospheric climate model |
| *RegCM* | Regional Climate Model |
| *WRF* | Weather Research and Forecasting model |
| **Land Surface Models** | |
| *CLM* | Community Land Surface Model |
| *iMOVE* | Interactive MOsaic-based VEgetation model |
| *NoahMP* | Noah Multi-Parameter model |

**Data availability**

The data and scripts used are available upon request from the corresponding author.

**Competing Interests**

The authors declare that they have no conflict of interest.

**Author Contributions**

PAM, DR, ELD, EK, MB, RMC, PH, DCAL, RM, PMMS, GS, SS, GS, MHT performed the RCM simulations, using vegetation maps produced by ELD. PAM designed the research, analysed the data, and wrote the paper. All authors contributed to interpreting the results and revising the text.

**Acknowledgements**

CICERO researchers acknowledge funding from the Research Council of Norway (grant 254966). In Norway, the simulations were stored on the server NIRD with resources provided by UNINETT Sigma2 - the National Infrastructure for High Performance Computing and Data Storage in Norway. WRFc-NoahMP simulations were performed and stored on resources provided by UNINETT Sigma2 - the National Infrastructure for High Performance Computing and Data Storage in Norway (NN9280K, NS9001K, NS9599K). WRFb-CLM4.0 simulations were supported by computational time granted from the National Infrastructures for Research and Technology S.A. (GRNET S.A.) in the National HPC facility - ARIS - under project ID pr005025 and pr007033_thin. Edouard L. Davin and Ronny Meier acknowledge financial support from the Swiss National Science Foundation (SNSF) through the CLIMPULSE project and thank the Swiss National Supercomputing Centre (CSCS) for providing computing resources. P. Hoffmann is funded by the Climate Service Center Germany (GERICS) of the Helmholtz-Zentrum Hereon in the framework of the Helmholtz-Institut Climate Service Science (HICSS) project LANDMATE. The authors gratefully acknowledge the WCRP CORDEX Flagship Pilot Study LUCAS "Land use and Climate Across Scales" and the research data exchange infrastructure and services provided by the Jülich Supercomputing Centre, Germany, as part of the Helmholtz Data Federation initiative. R. M. Cardoso, D. C. A. Lima P. M. M. Soares were supported by national funds through FCT (Fundação para a Ciência e a Tecnologia, Portugal) under project LEADING (PTDC/CTA-MET/28914/2017), and project UIDB/50019/2020. The research work from G. Sofiadis was supported by the Hellenic Foundation for Research and Innovation (HFRI) under the HFRI PhD Fellowship grant (Fellowship Number: 1359). P.A. Mooney was partially supported by the European Union's Horizon 2020 research and innovation framework programme under Grant agreement no. 101003590 (PolarRES).

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
