# Peer review of "Land-atmosphere interactions in sub-polar and alpine climates in the CORDEX FPS LUCAS models: II. The role of changing vegetation"

_The Cryosphere, 2021_

## Referee Comment (RC1)

Review of tc-2021-291
"Land-atmosphere interactions in sub-polar and alpine climates in the CORDEX FPS LUCAS models: II. The role of changing vegetation"
submitted to *The Cryosphere*

Synopsis:

This manuscript investigates the impact of land cover change on snow accumulation, ablation, and albedo. The authors use a nine-member regional climate model ensemble, with various combinations of atmospheric and land surface models. Using constant albedo differences between snow-covered and snow-free forest and grass, the authors calculate the snow albedo sensitivity index (SASI), using modeled snowpack variables as an indicator for snow-covered and snow-free conditions during the cold season (defined as January to June). The authors report that forested conditions reduce SASI, indicative of reduced climate forcing from snow albedo, with the greatest differences emerging during the ablation season.

Overall, I found this to be a clearly written and concise paper fit for publication in *The Cryosphere*. The figures are crisp, clean, and well organized.

Major Comments:

The authors used a constant $\Delta\alpha$ value of 0.5 for grass and 0.2 for forests (lines 138-140). Is there any reason the authors could not use the albedo values directly from the model output? I suspect albedo output was not available for all models? If this is the case, please indicate this in the text.

While I do not believe the results of the analysis would change (ie: SASI would still be reduced for forested conditions), use of model-derived $\Delta\alpha$ may shed additional light on why the models differ in their spatial and temporal evolution of the snowpack during the melt season. Use of modeled $\Delta\alpha$ may also illuminate differences in coniferous vs. deciduous forest effects on snow-vegetation albedo.

Minor comments:

Line 119:    Could the authors please provide a source justifying the use of 312 kg/m$^3$ for an average snow density?

Line 171:    Change km/m$^3$ to kg/m$^3$

Line 241:    Considering adding a few references on the implications for "cold and snow refugia" management strategies.
https://www.fs.usda.gov/ccrc/topics/climate-change-refugia

Figure 3 & 4:  Check units - SWE should be mm?

---

## Author Comment (AC2)

**Authors Response to Reviewer #2**

We greatly appreciate the positive and helpful feedback you have given us. Below, we respond to your comments point-by-point in blue for clarity.

**Major Comments**
This paper aims to analyze the impacts of afforestation/reforestation on snow and the snow-albedo effect and identify the major pathways to improve the model in representing grass-snow and forest-snow interactions. Based on a comparison between nine regional climate models with different combinations of regional atmosphere model and land surface scheme, the results show that there is large uncertainty in the magnitude for the changes in the snow-albedo sensitivity index, even though the sign of the change direction is robustly modeled by all the models. The greatest differences between models emerge in the snowmelt season, which is also seen in one regional model using different land surface models. In general, the manuscript touches upon very interesting scientific questions and has many potential merits on improving regional climate model or Earth system model to represent biophysical effects from land-use change or potential natural vegetation change. It falls well within the scope of the journal "the Cryosphere". However, this paper lacks detailed demonstration and interpretation of how different forests and grass regulates land surface albedo and energy balance. For vegetation-snow interaction, there are both positive and negative responses for forest displacement of grasslands to near-surface warming. A positive response means the decreased albedo enhances the net incoming shortwave radiation, while a negative response means the shading effects of taller woody species may delay the snowmelt in certain circumstances.

(1) the effects of afforestation (forest run – minus run) on SASI (spatially and seasonally)
We will add plots to our manuscript showing this.

(2) how the effects of afforestation on near-surface temperature, latent and sensible heat fluxes, and downward shortwave radiation
This analysis has been undertaken in a previous study by Davin et al. (2020), we will add text to the manuscript describing the results from Davin et al. (2020) and discuss our results and the results of Davin et al. (2020) together.

(3) why is the impact of afforestation (FOREST-GRASS) on the number of snow days in the season so different among the models, if they prescribe the same forest and grassland land cover
The difference between the models can be attributed to the model's representation of snow processes. Snow days depends on the variable snow depth which is closely related to snow water equivalent (SWE). In fact, some of the models derive snow depth from SWE. As highlighted in Thackeray et al. (2019) and Mudryk et al. (2020), there are known deficiencies in climate models ability to simulate snow mass variables such as snow water equivalent.

(4) is dynamic vegetation and static vegetation (prescribe phenology or no phenology) important for snow-albedo feedback?
This can be important (e.g. Cook et al. 2008). We will add text to the manuscript discussing this.

**Minor comments:**

1. Table 1.1. I hope the authors could mention more details about how snow-vegetation interaction is described by different land surface models.
   We will include additional information on snow-vegetation interactions in our manuscript.

2. Line 155. Do the deciduous and evergreen forests use the same albedo for free-snow surface and snow cover surface? These two types of forests have a big difference for the winter albedo.
   Deciduous and evergreen forests use the same albedo. Barlage et al. (2005) show that the values for snow covered evergreen needleleaf forests and deciduous broadleaf forest were 0.34 and 0.35, respectively. These differences are small and a recent study by Mooney et al. (2021) showed that the effect of evergreen needleleaf on afforestation in Norway was not significantly different from that of mixed forests which used the same albedo as deciduous forests. We will add text about this to the manuscript.

3. Figure 2. It would be nice to see the effects of afforestation by illustrating the difference between the forest run and grassland run.
   We will add plots to Figure 2 showing this.

4. Figure 3. How to explain the impact of afforestation (FOREST-GRASS) on snow water equivalent (SWE) also differs the sign among regional climate models.
   As discussed in our paper, it is widely known that climate models struggle to simulate snow water equivalent (SWE).

5. In the discussion part, it is worth mentioning the importance of surface roughness length and windblown snowdrift in the regional climate model to quantify snow-albedo effects of afforestation.
   This is a very good suggestion. We will certainly add some text on this important issue. Thank you for bringing it to our attention.

---

## Author Response (AR1)

**Authors Response to Reviewer #1**

Synopsis:
This manuscript investigates the impact of land cover change on snow accumulation, ablation, and albedo. The authors use a nine-member regional climate model ensemble, with various combinations of atmospheric and land surface models. Using constant albedo differences between snow-covered and snow-free forest and grass, the authors calculate the snow albedo sensitivity index (SASI), using modeled snowpack variables as an indicator for snow-covered and snow-free conditions during the cold season (defined as January to June). The authors report that forested conditions reduce SASI, indicative of reduced climate forcing from snow albedo, with the greatest differences emerging during the ablation season.
Overall, I found this to be a clearly written and concise paper fit for publication in *The Cryosphere*. The figures are crisp, clean, and well organized.

We greatly appreciate the positive and insightful feedback you have provided us. Below, we respond to your comments point-by-point (in blue to enhance clarity).

**Major Comments:**
The authors used a constant $\Delta\alpha$ value of 0.5 for grass and 0.2 for forests (lines 138-140). Is there any reason the authors could not use the albedo values directly from the model output? I suspect albedo output was not available for all models? If this is the case, please indicate this in the text.

While I do not believe the results of the analysis would change (ie: SASI would still be reduced for forested conditions), use of model-derived $\Delta\alpha$ may shed additional light on why the models differ in their spatial and temporal evolution of the snowpack during the melt season. Use of modeled $\Delta\alpha$ may also illuminate differences in coniferous vs. deciduous forest effects on snow-vegetation albedo.
The $\Delta\alpha$ values are the difference between snow covered and snow free albedo values. Constant values for $\Delta\alpha$ were used in this study as they could not be obtained from the model. We have indicated this in the text on line 176 on page 7.

**Minor comments:**
1. Line 119: Could the authors please provide a source justifying the use of 312 kg/m$^3$ for an average snow density?
   Indeed. We have cited the work of Sturm et al (2010) for this value.

   Sturm, M., Taras, B., Liston, G. E., Derksen, C., Jonas, T., & Lea, J. (2010). Estimating Snow Water Equivalent Using Snow Depth Data and Climate Classes, *Journal of Hydrometeorology*, *11*(6), 1380-1394. https://doi.org/10.1175/2010JHM1202.1

2. Line 171: Change km/m$^3$ to kg/m$^3$
   We have changed this.

3. Line 241: Considering adding a few references on the implications for "cold and snow refugia" management strategies. https://www.fs.usda.gov/ccrc/topics/climate-change-refugia

Thanks for bringing this to our attention. We have added the following text to section 4 (Discussion), lines 311-314.

"*Certain ecological species are also highly dependent on snow, and if afforestation mitigates some of the snow melt expected in a warmer world, then afforestation could help conservation efforts by offering potential sites for "climate change refugia" in a warmer world. Together these points, highlight the societal and ecological need for better information on the impact of afforestation in sub-polar and alpine regions, some of which are already undergoing afforestation.*"

4. Figure 3 & 4: Check units - SWE should be mm?

Thanks for pointing this out. This has been changed to mm.

**Authors Response to Reviewer #2**

We greatly appreciate the positive and helpful feedback you have given us. Below, we respond to your comments point-by-point in blue for clarity.

**Major Comments**

This paper aims to analyze the impacts of afforestation/reforestation on snow and the snow-albedo effect and identify the major pathways to improve the model in representing grass-snow and forest-snow interactions. Based on a comparison between nine regional climate models with different combinations of regional atmosphere model and land surface scheme, the results show that there is large uncertainty in the magnitude for the changes in the snow-albedo sensitivity index, even though the sign of the change direction is robustly modeled by all the models. The greatest differences between models emerge in the snowmelt season, which is also seen in one regional model using different land surface models. In general, the manuscript touches upon very interesting scientific questions and has many potential merits on improving regional climate model or Earth system model to represent biophysical effects from land-use change or potential natural vegetation change. It falls well within the scope of the journal "the Cryosphere". However, this paper lacks detailed demonstration and interpretation of how different forests and grass regulates land surface albedo and energy balance. For vegetation-snow interaction, there are both positive and negative responses for forest displacement of grasslands to near-surface warming. A positive response means the decreased albedo enhances the net incoming shortwave radiation, while a negative response means the shading effects of taller woody species may delay the snowmelt in certain circumstances.

(1) the effects of afforestation (forest run – minus run) on SASI (spatially and seasonally)

We have added plots to Figure 2 in our manuscript showing this.

(2) how the effects of afforestation on near-surface temperature, latent and sensible heat fluxes, and downward shortwave radiation

This analysis has been undertaken in a previous study by Davin et al. (2020). We have added the following text to section 3.1 SASI (Lines 201-206, page 8).

*"Figure 2 shows that GRASS simulations have higher SASI values than FOREST simulations meaning afforestation reduces the climate forcing from the snow-albedo effect. This is consistent with the findings of Davin et al. (2020) which also used the CORDEX FPS LUCAS models and showed that afforestation increased surface temperatures in all of these CORDEX FPS LUCAS models. Davin et al. (2020) also showed that afforestation increased net shortwave radiation and sensible heat considerably in these same models. This points to the decreased albedo from afforestation enhancing net shortwave radiation and leading to a positive response in surface air temperature to afforestation. In light of this, the lower SASI values for the FOREST simulations compared to the GRASS values can be primarily attributed to the difference in Δα which is 0.5 for GRASS and 0.2 for FOREST. Generally, afforestation does not impact the timing of the maximum value in SASI."*

(3) why is the impact of afforestation (FOREST-GRASS) on the number of snow days in the season so different among the models, if they prescribe the same forest and grassland land cover

The difference between the models can be attributed to the model's representation of snow processes. Snow days depends on the variable snow depth which is closely related to snow water equivalent (SWE). In fact, some of the models derive snow depth from SWE. As highlighted in Thackeray et al. (2019) and Mudryk et al. (2020), there are known deficiencies in climate models ability to simulate snow mass variables such as snow water equivalent.

(4) is dynamic vegetation and static vegetation (prescribe phenology or no phenology) important for snow-albedo feedback?

This can be important (e.g. Cook et al. 2008). We added the following text to the conclusions (Lines 356-360, page 16).

*"Future studies could also consider the impact of dynamic vegetation modelling on the snow-albedo feedback. Previous studies such as Cook et al. (2008) have shown that dynamic vegetation in climate models can be an important amplifier of the snow-albedo feedback. Such analysis was not possible in this study as most models did not have this capability. Future studies should examine this when more land surface models have developed this capacity."*

**Minor comments:**
1. Table 1.1. I hope the authors could mention more details about how snow-vegetation interaction is described by different land surface models.

   We included additional information on snow-vegetation interactions in our manuscript in a new sub-section 2.2.1. (Lines 122-160, pages 5 and 6).

2. Line 155. Do the deciduous and evergreen forests use the same albedo for free-snow surface and snow cover surface? These two types of forests have a big difference for the winter albedo.

   Deciduous and evergreen forests use the same albedo. Barlage et al. (2005) show that the values for snow covered evergreen needleleaf forests and deciduous broadleaf forest were 0.34 and 0.35, respectively. These differences are small and a recent study by Mooney et al. (2021) showed that the effect of evergreen needleleaf on afforestation in Norway was not significantly different from that of mixed forests which used the same albedo as deciduous forests. We added the following text to section 2.3 (Lines 178-183, page 7):

*"Although there are differences in the snow-covered values for different types of forests, i.e., deciduous vs evergreen, these differences were very small (0.35 vs 0.34). Previous studies on the impacts of afforestation in sub-polar climates, such as Mooney et al. (2021), have shown that such small changes in albedo have a negligible impact on the outcome. Based on this, the same value was used for all forests regardless of forest type."*

3. Figure 2. It would be nice to see the effects of afforestation by illustrating the difference between the forest run and grassland run.
   We added plots to Figure 2 showing this.

4. Figure 3. How to explain the impact of afforestation (FOREST-GRASS) on snow water equivalent (SWE) also differs the sign among regional climate models.
   As discussed in our paper, it is widely known that climate models struggle to simulate snow water equivalent (SWE).

5. In the discussion part, it is worth mentioning the importance of surface roughness length and windblown snowdrift in the regional climate model to quantify snow-albedo effects of afforestation.
   This is a very good suggestion. Thank you for bringing it to our attention. We added the following text to the conclusions in section 5 (Lines 431-343, page 16).
   *"A known deficiency in regional models is their inability to represent windblown snowdrift, which is an important factor for quantifying the snow-albedo effects of afforestation."*